# Feasibility of an At-Home Adult Stool Specimen Collection Method in Rural Cambodia

**DOI:** 10.3390/ijerph182312430

**Published:** 2021-11-26

**Authors:** Jordie A. J. Fischer, Crystal D. Karakochuk

**Affiliations:** 1Food, Nutrition, and Health, The University of British Columbia, Vancouver, BC V6T 1Z4, Canada; jordie.fischer@ubc.ca; 2Healthy Starts, BC Children’s Hospital Research Institute, Vancouver, BC V5Z 4H4, Canada

**Keywords:** stool, stool collection, Cambodia, sample collection, rural health, gut microbiome, collection kit

## Abstract

The human microbiome has received significant attention over the past decade regarding its potential impact on health. Epidemiological and intervention studies often rely on at-home stool collection methods designed for high-resource settings, such as access to an improved toilet with a modern toilet seat. However, this is not always appropriate or applicable to low-resource settings. Therefore, the design of a user-friendly stool collection kit for low-resource rural settings is needed. We describe the development, assembly, and user experience of a simple and low-cost at-home stool collection kit for women living in rural Cambodia as part of a randomized controlled trial in 2020. Participants were provided with the stool collection kit and detailed verbal instruction. Enrolled women (*n* = 480) provided two stool specimens (at the start of the trial and after 12 weeks) at their home and brought them to the health centre that morning in a sterile collection container. User specimen collection compliance was high, with 90% (*n* = 434) of women providing a stool specimen at the end of the trial (after 12 weeks). This feasible and straightforward method has strong potential for similar or adapted use among adults residing in other rural or low-resource contexts.

## 1. Introduction

There has been a growing interest in understanding the composition of the human gut microbiome and its linkages with all areas of health [1]. This has led to increased microbial community analysis, using techniques including 16 s rRNA gene sequencing, quantitative PCR, and culture-based methods [2,3]. Increasingly, large-scale observational and intervention studies have aimed to collect stool specimens to provide data in this rapidly evolving field of microbiology.

In both clinical practice and research studies, it has become increasingly common for stool specimen collection to be completed at an individual’s home and then shipped to the laboratory [4,5]. At-home stool collection kits are often designed for a modern toilet seat and depend on a reliable national priority mail delivery system, [5,6] such as the widely used OMNIgene•GUT (DNA Genotek, Ottawa, ON, Canada) along with the OM-AC1 toilet accessory, a flushable collection paper secured to the toilet seat with adhesive strips [7].

Nevertheless, there is limited data concerning practical methods for collecting stool specimens for microbiome analyses in settings outside of the high-resource contexts, specifically for those without modern toilet seats or reliable shipment options. Thus, in an effort to characterize the human microbiome across the full range of the human experience, populations in low-resource settings continue to be underrepresented [8]. In the context of large studies in low-resource settings, specimen transportation and refrigeration, contamination, and acceptable collection may pose challenges. Designing a simple, user-friendly stool specimen collection kit for use in these challenging environments is imperative.

Here, we report on study tools developed and used to collect neat stool specimens from women in rural Cambodia, used as a part of a randomized controlled trial that aimed to evaluate the potential harms of iron supplementation [9]. We discuss the feasibility and acceptability of our convenient at-home stool collection methods that have the potential to be implemented in similar low-resource locations. 

## 2. Materials and Methods

### 2.1. Study Participants and Setting

This stool specimen collection methodology was designed as a part of a randomized controlled trial in rural Kampong Thom province, Cambodia, with ethics granted from the Clinical Research Ethics Board at the University of British Columbia in Vancouver, Canada (H18-02610), and the National Ethics Committee for Health Research in Phnom Penh, Cambodia (273-NECHR). All participants provided written informed consent before participating in the study, including the collection of baseline and 12-week venous blood, neat stool, and fecal swab specimen. Full details of the original study can be found elsewhere [9], and the trial was registered at clinicaltrials.gov (NCT04017598). Between December 2019 and January 2020, women were recruited, and *n* = 480 non-pregnant women of reproductive age (18–45 years) were randomized to 12 weeks of daily oral iron supplementation in the form of either 60 mg ferrous sulfate (*n* = 161), 18 mg ferrous bisglycinate (*n* = 158), or placebo (*n* = 161). Venous blood specimens and neat stool specimens were collected at baseline and after 12 weeks.

Enrolled women did not have access to modern toilets with toilet seats nor household refrigeration. The most common type of sanitation facility available at most households was pour-flush to a septic tank or pit latrines (91% [437/480]).

### 2.2. Stool Collection Protocol

Development of the stool collection methodology took place in discussion with local public health staff highly experienced in rural specimen collection and knowledgeable about resource limitations of the study location (i.e., toilet facilities). Research staff were trained on the use of the stool collection kits and the procedures to disperse and collect the specimens from research participants. At the initial study visit, following administration of the baseline questionnaire, study staff provided the participants with the stool collection kit, verbally explained how to properly collect the specimen in Khmer (local Cambodian language), and dispose of collection materials. Women were also provided with a written copy of the same instructions regarding stool collection to take home via a simple Khmer-translated infographic. They were instructed to bring their stool specimen back to the local health centre the following day. 

The stool collection kit was labelled with the participant ID number. It contained the following items: 30 mL clear polystyrene stool collection container with a screw-on blue lid and attached spoon, gloves, Khmer translated infographics and a metal pot (Figure 1). The metal pots were stored in heavy-duty plastic bags to prevent contamination before distribution to participants.

The verbal instructions as provided in staff training and written infographic (Figure 2) were communicated to study participants as follows:Collect first stool the morning of your health centre visit.Put on gloves provided in the stool collection bag.Squat or sit over the provided metal pot.Ensure that the pot is not touching toilet water—make sure no water, other liquids or materials get into the pot.Defecate into the pot. A small amount of stool is ok.Open the stool container tube by unscrewing the blue lid.Use scoop attached to blue lib to collect a small portion of stool from the pot (size of cashew nut).Place stool specimen and scoop into the stool collection tube and screw tight to secure lid.Place the tube into the stool collection bag with your personal ID number.Dispose of or clean the metal pot thoroughly with soap and hot water.Thoroughly wash hands with soap and hot water.Bring stool specimen to study nurse at the health centre on the same morning.

Women collected their stool specimens at their home and brought the completed kit back to the health center within ~2 h of defecation in the provided clear resealable plastic bag. Many participants opted to place the transparent bag inside a small opaque black garbage bag for additional privacy. Upon retrieval, study staff would ensure the stool collection kit contained the neat stool specimen and the container was tightly sealed. Tubes were labelled with the participant, study visit number, date, and time received. Labelled tubes were double-checked to ensure participant ID matched the ID number marked on the outer side of the bag. The kits were then immediately placed on ice. Neat stool specimens were transported on ice to the National Public Health Laboratory, where specimens underwent further processing and were frozen at −20 °C within 4–6 h until further analysis. Additionally, women were given the metal pot to keep and use for additional study visits where follow-up stool collection was needed. 

Missing stool specimens were documented, and women were followed up by staff on the morning of the initial study visit. If a woman could not pass stool or was not available for a study visit that day, stool specimens were collected within seven days of the original study visit date. In this event, study staff called women to arrange another stool pickup, ensuring that pickup happened within 2 h of bowel movement and placed on ice, driven to the National laboratory and frozen at −20 °C within 4–6 h. 

## 3. Results

During recruitment, *n* = 1286 women were screened for study inclusion eligibility, of which *n* = 577 did not meet the inclusion criteria, and *n* = 229 declined to participate. No women declined to participate due to the requirement of a stool specimen collection. A total of 480 women were enrolled in the study, and *n* = 456 (95%) women provided a stool specimen at baseline (baseline blood collection, not stool collection, was required for enrollment). A total of *n* = 441 (92%) women remained in the study until completion at 12 weeks, with *n* = 434 (90%) women providing a 12-week stool specimen, depicted in Figure 3. Women who provided a baseline stool specimen (*n* = 456) and those that did not give a baseline stool specimen (*n* = 22) differed by education level achieved (fisher’s exact, *p* = 0.044), breastfeeding status (fisher’s exact, *p* = 0.007), reported diarrhea (3+ loose bowel movements in 24 h) (fisher’s exact, *p* = 0.023), pain when passing stool (fisher’s exact, *p* = 0.015), blood in stool (fisher’s exact, *p* = 0.028), and if women had previously taken antibiotics (fisher’s exact, *p* = 0.006) (Table 1).

Throughout the study duration, research staff informally collected feedback from participants on their experience in the stool collection method. Some specific comments from research participants included: constipation made stool collection a challenge, this was their first time providing a stool specimen, and lastly, in general, participants expressed greater hesitation and fear towards blood collection than stool collection. It should be emphasized that this data was not systematically collected and may be biased by many factors, such as response bias. Research staff shared that verbal communication was more productive and helpful to participants than the written instructions (Khmer translated infographic). 

DNA was extracted from a subsample (*n* = 150) of thawed neat stool using QIAamp PowerFecal Pro DNA Kit [10], and DNA yield was checked via NanoDrop spectrophotometer reading. All extracted specimens yielded DNA suitable for PCR amplification and were thus uncompromised during specimen collection, transportation, and storage. 

## 4. Discussion

With the call and opportunity to promote inclusivity in microbiome research, an appropriate, low-cost and appropriate stool collection method is warranted for use in rural and low-resource settings [8,11]. The collection of stool specimens from a large cohort of rural-dwelling women could present challenges regarding participant recruitment, specimen collection, retention, and management of staff resources. We describe the development, assembly and use of a simple, low-cost at-home stool collection kit for rural or low-resource settings where modern toilets with seats are unavailable. Using this at-home stool collection kit was reported as easy and safe. 

Other authors have reported using at-home adult stool collection tools but are also limited by the availability of a modern toilet with a toilet seat [5,6]. Further, contemporary over-the-toilet seat stool collection supplies (e.g., flushable collection paper secured to the toilet seat) cannot be procured in some countries, such as Cambodia, and instructions are not available in different languages. In rural and low-resource settings, even when improved sanitation infrastructure is built, there is still a lack of facilities allowing for straightforward and sterile stool collection. Our kit was <$5 USD, thus, it is a low-cost option for single and multiple follow-up stool specimen collections. 

To our knowledge, no authors have reported on the development and use of a simple adult stool collection kit for use at an individual’s home in a rural or low-resource setting. There is no consensus or guidance on an appropriate or detailed method of stool collection in rural or low-resource settings. In other reports in rural and low-resource settings, the general practice is to collect the stool specimen at the local health centre [12], which may be unfeasible for studies with large sample sizes and for women who cannot defecate ‘on demand’. Our stool collection method is novel, as it allows participants to independently collect a stool specimen in the comfort and privacy of their home and when they feel the ‘urge’ to defecate. This approach also reduces the burden on local health facilities and research staff and is optimal for use in large-scale research.

Ensuring that specimen collection methods are culturally acceptable is essential to improve participation and minimize attrition rates in the study population. On account of our high study retention rate (92%) and stool specimen collection rate at 12 weeks (90%), we infer participants generally accepted this stool specimen collection method. However, these findings may have been affected by response bias. We should reiterate that our high study retention rate was likely due to our experienced field research staff’s strong rapport with study participants. Detailed staff training resulted in the clear communication of stool collection instructions. We also recognize the limitations of this method of stool specimen collection. Although the metal pots were stored together in heavy-duty plastic bags to prevent contamination by dirt and debris, they were not stored in a sanitized environment, allowing for possible contamination during storage and transportation. As a lesson learned, we recommend that collection pots/containers be wrapped in protective sealing and stored in clean areas. Alternatively, the collection pots/containers could be sanitized prior to defecation at each specimen collection time point, if participants were provided with such materials. Secondly, our research group provided clear bags for the transportation of stool specimens to the health centre. Still, most participants opted to put the clear participant labelled bag inside in their own small black garbage bag for privacy. Therefore, we recommend supplying a discrete, non-opaque bag or container for participants to return specimens to the study staff. Lastly, it would be advantageous to conduct a standardized assessment of user acceptability of this stool specimen collection technique in future work.

## 5. Conclusions

In response to the growing field of human gut microbiome study, we describe the development, assembly and use of a simple, low-cost at-home stool collection kit for rural or low-resource settings where modern toilets with seats are unavailable. This method for the collection of stool specimens was feasible, generally acceptable, and has strong potential for similar or adapted stool collection procedures among adults residing in other rural or low-resource settings. 

## Figures and Tables

**Figure 1 ijerph-18-12430-f001:**
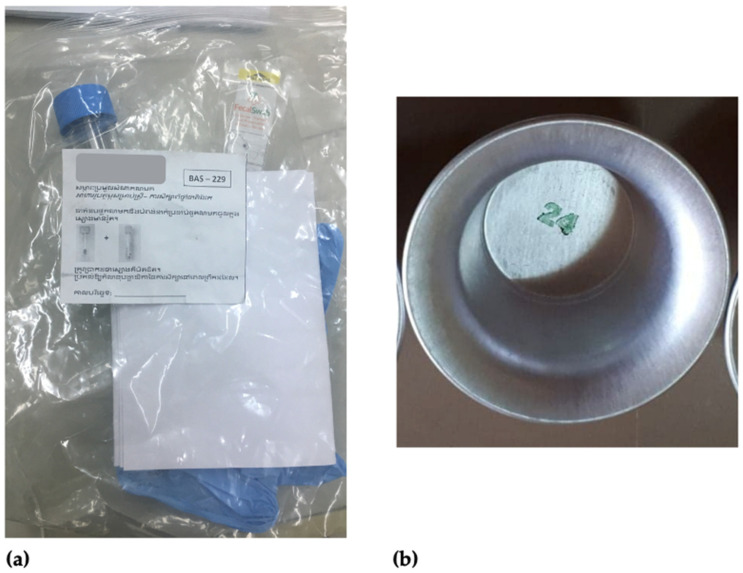
Stool Collection Kit: (**a**) Resealable participant labelled bag, infographic, gloves, 30 mL clear polystyrene stool collection container; (**b**) Metal collection pot.

**Figure 2 ijerph-18-12430-f002:**
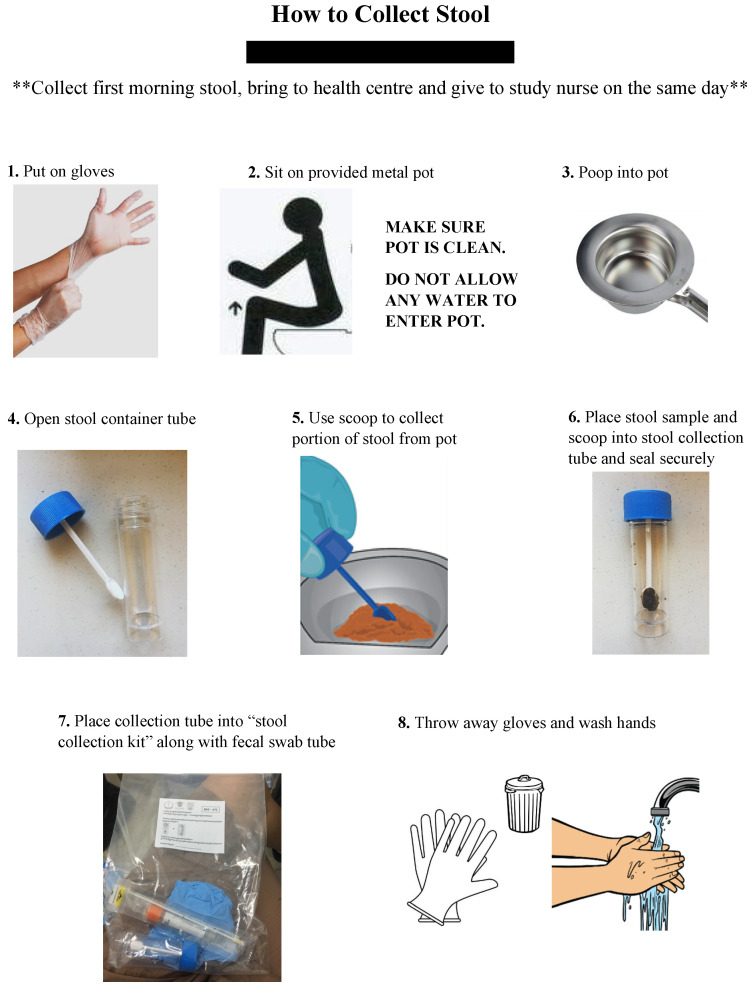
Participant stool collection infographic (English translation).

**Figure 3 ijerph-18-12430-f003:**
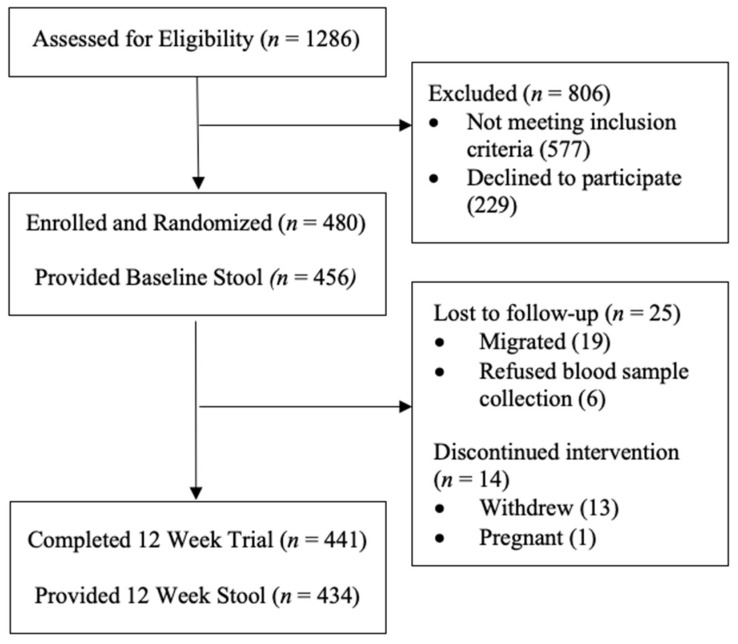
Participant flow chart.

**Table 1 ijerph-18-12430-t001:** Baseline participant characteristics of enrolled Cambodian women by provision of baseline stool specimen.

	Provided Baseline Stool Specimen	No Baseline Stool Specimen	*p*-Value ^1^
Total enrolled, *n* (%)	458 (95%)	22 (5%)	
Woman’s age, y, median (IQR)	34.5 (28.0, 40.0)	31.5 (29.0, 36.0)	0.387
Married	397/458 (87%)	19/22 (86%)	0.624
Completed education, (%)			0.044 *
Primary	242/416 (58%)	9/22 (41%)	
Lower secondary	106/416 (26%)	12/22 (55%)	
Upper secondary	54/416 (13%)	1/22 (4%)	
Higher education/university	14/416 (3%)	0/22 (0%)	
BMI (kg/m^2^)	23.5 ± 3.8	22.7 ± 3.6	0.295
Currently breastfeeding	40/141 (28%)	43/145 (30%)	0.007 *
Currently use birth control	56/161 (35%)	70/158 (44%)	0.826
Previously taken antibiotics	202/458 (44%)	11/22 (50%)	0.006 *
Experienced gastrointestinal upset			
Diarrhea	67/241 (28%)	0/13 (0%)	0.023 *
Nausea	126/241 (52%)	7/13 (54%)	1.00
Constipation	31/241 (13%)	3/13 (23%)	0.392
Pain when passing stool	71/241 (7%)	4/13 (31%)	0.015 *
Blood in Stool	11/241 (5%)	3/13 (23%)	0.028 *

Total *n =* 480. Values are *n* (%) or median (IQR). ^1^ Independent samples *t*-test (parametric) and Wilcoxon rank sum tests (non-parametric) for continuous variables and Fisher’s exact tests for categorical variables. * Statistically significant at *p* < 0.05.

## Data Availability

All relevant data are within the manuscript.

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
