# Peer review of "Feasibility of an At-Home Adult Stool Specimen Collection Method in Rural Cambodia"

_ijerph, 2021, doi:10.3390/ijerph182312430_

Round 1

Reviewer 1 Report

The objective of the study was to create a low cost, user friendly stool collection kit to be used in  studies in regions with limited resources.  Data collected by the authors overall demonstrated the feasibility and acceptance of such kits by the women who were enrolled in study.  

Minor comments:

  1. There was an inconsistency of the number of women who provided baseline stool samples. Fig3 indicated 456 women provided baseline stool samples but line 140 indicated that 458 women provided baseline stool samples. 
  2. The authors described that the cost for a single use of the kit was less than $5, however, for 12- weeks collection, the cost could add up very significantly. This limitation should be discussed.

Author Response

  1. There was an inconsistency of the number of women who provided baseline stool samples. Fig3 indicated 456 women provided baseline stool samples but line 140 indicated that 458 women provided baseline stool samples.  

Response 1: Thank you for noticing this inconsistency. Line 140 has been corrected to 456 women.

  1. The authors described that the cost for a single use of the kit was less than $5, however, for 12- weeks collection, the cost could add up very significantly. This limitation should be discussed.

Response 2: For our study, we only collected stool at two time points: week 0 and week 12. Therefore, costs remained less than $5, but stool collection costs would increase if more collection time points were included. Nevertheless, this remains a low-cost stool collection kit option. 

Reviewer 2 Report

The Authors present an interesting article entitled “Feasibility of an at-home adult stool specimen collection method in rural Cambodia”.

The topic is definitely of interest for the community because it explore a particular reality of the world and it might be helpful in moving it forward.
This is a very well written manuscript and correct in the methodology of the research..

the strength of the manuscript concerns a study carried out in a rural reality about which little is known to the scientific community. Although it concerns an aspect that may seem basic, the effort made the idea, design and implementation of the experimenta work are factors that provide a great value to this article, in fact, the strategy proposed may be useful for other similar low-resouces settings.
It is appreciable that the same Authors describe the limits of the work which, however, are related to technical issues concerning the stool specimen collection. 
I think this manuscript can be accepted, precisely because it is important that scientific literature and in particular an editor like MDPI that has a great impact on it, describe minor and little known realities but of equal value for the world community.

Author Response

There are no questions or comments to respond to. Thank you for your positive feedback.

Reviewer 3 Report

Thank you for allowing me to review your manuscript. 

Please see my comments below:

  1. Line 84 details what was included in the stool collection kit that contains one polystyrene collection container. But lines 116 - 121 explains that the study staff would ensure the collection kit contained "both the neat stool specimen and the swab and that both were tightly sealed". Where did the swab come from for the women to use it and what was it used for as this is not explained in the manuscript? The neat stool went to the National Public Health Laboratory, but where did the swab go?
  2. As the authors acknowledged, the steel pot was not sterile and its cleanliness was unknown which could have a large impact on stool microbiome data analysis. The recommendation for sterilisation prior to use would be beneficial especially if the pots are being use on multiple occasions, they would need sterilisation before each use.
  3. It was acknowledged that participant feedback was informally collected which is a shame as it would be beneficial to have reliable feedback using this method for stool collection to get a better idea for acceptability and retention of participants in low-resource areas when they potentially won't have the support from local field research staff as you did with this cohort. 

Author Response

  1. Line 84 details what was included in the stool collection kit that contains one polystyrene collection container. But lines 116 - 121 explains that the study staff would ensure the collection kit contained "both the neat stool specimen and the swab and that both were tightly sealed". Where did the swab come from for the women to use it and what was it used for as this is not explained in the manuscript? The neat stool went to the National Public Health Laboratory, but where did the swab go?

Response 1: The fecal swab was also included in the stool collection kit (please see the infographic with a photo of kit). Reading your question, we agree it is confusing and not relevant to mention the fecal swab. The swab was used to answer the research questions of the larger trial that this sub-study was nested in. The swab does not address the research objective of this sub-study to create a low-cost, user-friendly stool collection kit to be used in studies in regions with limited resources. We have removed the mention of the fecal swab from lines 116-121. 

The text now reads: “Upon retrieval, study staff would ensure the stool collection kit contained the neat stool specimen and the container was tightly sealed. Tubes were labelled with the participant, study visit number, date, and time received. Labelled tubes were double-checked to ensure participant ID matched the ID number marked on the outer side of the bag. The kits were then immediately placed on ice. Neat stool specimens were transported on ice to the National Public Health Laboratory…”

2. As the authors acknowledged, the steel pot was not sterile, and its cleanliness was unknown which could have a large impact on stool microbiome data analysis. The recommendation for sterilization prior to use would be beneficial especially if the pots are being used on multiple occasions, they would need sterilization before each use. 

Response 2: We agree it is vital that the collection container must be sanitized before each specimen collection time point. In response to your comment, we have added additional text on line 212 to reiterate this point in the discussion question. 

Lines 211-213 now read: “Alternatively, the collection pots/containers could be sanitized prior to defecation at each specimen collection time point if participants were provided with such materials.”

3. It was acknowledged that participant feedback was informally collected which is a shame as it would be beneficial to have reliable feedback using this method for stool collection to get a better idea for acceptability and retention of participants in low-resource areas when they potentially won't have the support from local field research staff as you did with this cohort. 

Response 3: We agree; it is too bad this data was not formally collected. We state there is an opportunity for future work in this area on lines 217-219 that reads: “Lastly, it would be advantageous to conduct a standardized assessment of user acceptability of this stool specimen collection technique in future work.”